# Inter-lower limb and intra-lower limb muscle activity correlations during walking: A comparative study of stroke patients and healthy individuals

**Thanita Sanghan**[1], **Nusreena Hohsoh**[1], **Goran Stojanović**[2], **Rezaul Begg**[3], **Surapong Chatpun**[1*]

1 Department of Biomedical Sciences and Biomedical Engineering, Faculty of Medicine, Prince of Songkla University, Songkhla, Thailand, 2 Department of Electronics, Faculty of Technical Sciences, University of Novi Sad, Novi Sad, Serbia, 3 Institute for Health and Sport, Victoria University, Melbourne, Victoria, Australia

\* surapong.c@psu.ac.th

## Abstract

Gait analysis is a crucial tool for understanding human locomotion, particularly in clinical settings where it aids in diagnosing and managing conditions that affect movement. This study investigated muscle activity, intra-, and inter-muscular correlations during gait phases, including first double support (DS1), single support (SS), second double support (DS2), and swing (SW) in both healthy individuals and stroke patients. By examining the root mean square (RMS) values and the area under the curve of RMS envelope of EMG for the gluteus medius (GM), rectus femoris (RF), biceps femoris (BF), medial gastrocnemius (MG), tibialis anterior (TA) and peroneus longus (PL) muscles, this research was designed to enhance our understanding of muscle coordination and its complications in stroke gait rehabilitation. The findings indicated significant differences in muscular activity and inter-limb correlations between limbs within both groups. Stroke patients exhibited knee hyperextension during stance phase, primarily due to muscle weakness and compensatory mechanisms, which may coexist with stiff knee gait during swing phase. Healthy participants showed proximal muscle weakness, likely age-related. Asymmetries in muscle activation between limbs were also observed in both groups, contributing to gait instability and an increased risk of falls. This study provides insights into the differences in muscle coordination between healthy individuals and stroke patients during various phases of walking. The findings emphasize the importance of training distal muscles, including the MG, TA, and PL muscles, in stroke patients, particularly in elderly, while also focusing on core/proximal muscles like the GM, RF, and BF muscles to improve weight-bearing support and gait stability, especially in cases of limited ankle movement. For age-induced muscle weakness, the emphasis should be on distal muscles.

**Data availability statement:** All EMG measuring files are available from the URL https://doi.org/10.6084/m9.figshare.28251239.v1.

**Funding:** This study was funded by the Faculty of Medicine, Prince of Songkla University, Thailand, 64-386-25-2, (SC), https://medinfo.psu.ac.th/home/ and the European Union's Horizon Europe Research and Innovation Programme, 101086348, (GS), https://research-and-innovation.ec.europa.eu/funding/funding-opportunities/funding-programmes-and-open-calls/horizon-europe_en. The funders did not play any role in the study design, data collection and analysis, decision to publish, or preparation of the manuscript.

**Competing interests:** The authors have declared that no competing interests exist.

These findings have the potential to improve clinical practices by offering better-tailored interventions to enhance mobility and quality of life in stroke survivors.

## Introduction

Stroke ranks as the third leading cause of death and disability grobally, with its incidence rising each year. Over 100 million individuals worldwide have experienced stroke and continue to live with its consequences [1,2]. As a major public health concern, stroke often results in motor impairments that impact daily activities such as walking and other basic activities in daily life. Understanding the complexities of gait patterns in stroke patients is essential for developing effective rehabilitation strategies [3]. Recently, the study of gait in stroke patients has gained significant attention due to the significant impact on motor function associated with muscle weakness, impaired coordination, and altered muscle activation patterns, specifically affecting walking abilities and mobility [4]. Several studies compared healthy subjects and stroke patients during walking and reported that the asymmetry and variability indices of spatiotemporal parameters can indicate the gait abnormalities [5,6].

In healthy individuals, walking requires the coordinated effort of various muscles to ensure smooth and efficient movement through the different phases of the gait cycle. In contrast, stroke patients typically reveal asymmetry in muscle activity and coordination between the affected and non-affected limbs [7] and the analysis of muscle synergies is a vital tool for assessing motor impairment and evaluating the effectiveness of rehabilitation [8]. Gait impairments in stroke patients are typically characterized by asymmetry and slow walking speed, often due to motor weakness, motor control deficit, sensory loss, proprioceptive issues, or ataxia [9,10]. In addition, muscle dysfunction contributes to gait problems such as stiff knee gait, knee hyperextension, and foot drop. These impairments combine with compensatory strategies like circumduction or vaulting, further disrupt foot clearance and reduce walking efficiency [11].

Despite extensive research on gait analysis in both healthy individuals and stroke patients, there remain gaps in our understanding of the complex dynamics of muscle activity during walking. Most existing studies either focussed on overall gait patterns or isolated muscle activation patterns, without thoroughly exploring inter-muscular correlations throughout the gait cycle. Furthermore, while it is known that stroke leads to asymmetries in muscle activation, the extent and implications of these asymmetries on intra- and inter-limb coordination are not fully understood. Comprehensive studies are, therefore, needed to investigate the relationships between muscle groups and their collective impact on gait stability and efficiency in stroke patients.

This study aimed to fill research gaps by providing a detailed analysis of muscle activity, intra-, and inter-muscular correlations for lower limbs during the gait cycle in both healthy individuals and stroke patients. By comparing the RMS values and the area under the curve of RMS of EMG envelope across different gait phases, we

sought to enhance understanding of muscle coordination and its impact on gait performance, with potential implications for stroke rehabilitation.

## Methods

### Subjects

The participants were fifteen stroke patients and fifteen healthy individuals recruited from Songklanagarind hospital from 30 November 2021 to 14 August 2022. The stroke patients were over 45 years with first-onset stroke, medically stable and able to walk without assistance under supervision of physiotherapist and able to follow the step command. Stroke patients with a prior history of musculoskeletal conditions unrelated to stroke for instance acute low back pain within 6 weeks prior to participation, other neurological conditions (e.g., Parkinson's disease, apraxia, seizure), lower limb arthroplasty, major falls, and fear of falling were excluded. The control group were healthy individuals of the same age range who could continue daily activity without angina chest pain or falling. Participants signed the informed consent prior to participating in this study. The demographics of the participants are summarized in Table 1.

### Study protocol

Ethical approval was obtained from the human research ethics unit under the Faculty of Medicine, Prince of Songkla University (REC.64-386-25-2). The participants were instructed to complete three sets of two-round walking at a comfortable speed along 10 meter-walkway (Fig 1). Then, a 5-minute rest was provided between each set to avoid fatigue that could impact their walking pattern. All gait data were gathered using the Pedar system (Novel, Germany) and Vicon motion capture system (Vicon Motion Systems Ltd, United Kingdom). Pedar, equipped with 99 sensors per insole, is an accurate and reliable in-shoe dynamic plantar pressure measuring apparatus. The Vicon system, recognized as the gold standard motion capture technology, was employed for tracking muscle activity through the Zerowire EMG system. The muscles of interest included Gluteus medius (GM), Rectus femoris (RF), Biceps femoris (BF), Medial gastrocnemius (MG), Tibialis anterior (TA), and Peroneus longus (PL).

**Table 1. Characteristics of participants.**

|  | Stroke patients (n = 15) | Healthy subjects (n = 15) |
|---|---|---|
| Age (years) | 60.87 (6.99) | 59.60 (6.32) |
| Sex (n): male/female | 11/4 | 9/6 |
| Height (m) | 1.65 (0.08) | 1.60 (0.10) |
| Weight (kg) | 67.93 (11.87) | 61.63 (11.77) |
| BMI (kg/m²) | 24.83 (3.51) | 23.91 (2.84) |
| MoCA score (Point) | 20.85 (6.01) | 25.67 (2.41) |
| Stroke onset (years) | 4.06 (3.78) | – |
| Affected side (R/L) | 6/9 | – |
| Walking speed (m/s) (Left-affected/Right-affected) | 0.37 (0.17)/0.61 (0.33) | 1.17 (0.21) |
| Manual muscle test (Affected/Non-affected) |  |  |
| Hip flexor | 6.91 (1.51)/9.64 (0.81) | – |
| Hip extensor | 7.18 (1.40)/9.64 (0.81) | – |
| Knee flexor | 5.64 (2.69)/9.82 (0.60) | – |
| Knee extensor | 6.18 (2.18)/9.82 (0.60) | – |
| Ankle dorsiflexor | 3.45 (2.42)/9.64 (0.81) | – |
| Ankle plantarflexor | 3.73 (2.76)/9.64 (0.81) | – |

Values are presented as mean (SD).

For electromyography (EMG) measurements, the skin of individuals was shaved and cleaned with an alcohol pad to reduce skin impedance. Subsequently, Ag/AgCl electrodes were placed at the motor point of the targeted muscles, following the specific guidelines outlined in the SENIAM protocol [12]. To initiate the plantar pressure data collection process, the subjects were instructed to offload the foot from the insole on the left and right sides.

## Data processing

Gait assessment was conducted with a sampling rate of 100 Hz for the Pedar system and 1000 Hz for the Vicon. Gait variables obtained and analyzed included ground reaction force (GRF) and muscle activity measured via EMG. All EMG data analyses were conducted using MATLAB 2020b. The collected EMG data underwent filtering using a fourth-order Butterworth bandpass filter with cutoff frequency of 30–450 Hz and a 50 Hz notch filter [13,14]. Subsequently, the filtered EMG data were synchronized with the GRF data and interpolation was performed to estimate missing data points due to differences in sampling rates. Normalization of the EMG data was performed based on the mean value of the EMG data. Four sets of five consecutive gait cycles, twenty gait cycles for each participant, were included in the analysis. Then, each gait cycle was analysed using gait phases defined by events from the GRF data as shown in Fig 2.

The gait phases include the first double support (DS1), single support (SS), second double support (DS2), and swing (SW) phases [15]. The duration was determined as the average percentage of total gait cycle time over each limb from both groups. In order to determine the difference between both limbs within each group, the root mean square (RMS) was used to quantify the overall magnitude of muscle activation, as calculated using Equation (1).

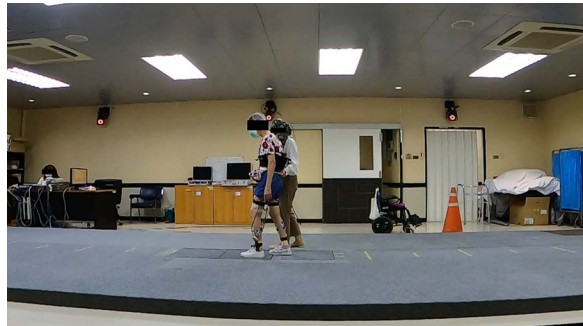

**Fig 1. Experimental setup of stroke patient with attached devices during examination.**

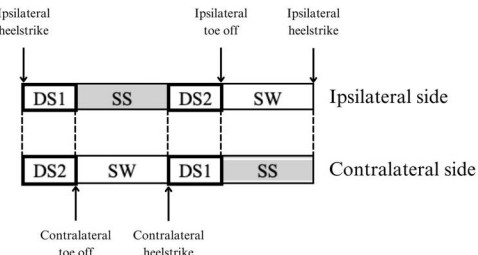

**Fig 2. Gait subphase over one gait cycle which consists of first double support (DS1), single support (SS), second double support (DS2) and swing (SW) phases.**

$$RMS = \sqrt{\frac{1}{n} \sum_{i=1}^{n} x_i^2}$$

(1)

To determine the gait phase based on changes in the amplitude and timing of muscle activity, the envelope of an EMG signal can be obtained by computing the local RMS value of the raw EMG via a 50-ms moving window[16]. The area under the RMS envelope curve was then calculated to compare EMG activation between limbs, serving as a measure of voluntary muscle drive [17].

### Statistical analysis

Statistical analysis was performed using Prism9 (GraphPad Software, MA, USA) and R software for statistical graphics (RStudio, MA, USA). The Shapiro-Wilk test was employed to assess the normality of distribution. The Wilcoxon matched-pairs signed rank test was used to verify the statistical difference between limbs in both groups. The significance levels were set at $p < 0.05$.

To explore the relationship between muscle activities, the non-parametric Spearman correlation were utilized to determine intra-limb and inter-limb comparision. Spearman correlation coefficient range is -1–1 with the sign of the Spearman correlation indicating the direction of association between the independent variable (x-axis) and the dependent variable (y-axis), with a value less than 0.40 indicating a weak correlation. Correlation coefficients ranging from 0.40 to 0.69 suggested a moderate correlation, while values between 0.70 and 0.89 are interpreted as a strong correlation between variables. A range from 0.90 to 1.00 suggests a very strong correlation [18].

### Results

#### Comparison of the RMS and the area under the curve of RMS envelope of EMG

Tables 2 and 3 present a comparison of the RMS values for each muscle and the area under the curve of RMS envelope of EMG throughout the gait cycle between healthy participants and each affected-side group of stroke patients, respectively. In the healthy group, significant differences in RMS values between left and right sides were observed in the GM, RF, and PL during the DS1 phase; in the BF during the SS phase; and in the RF during the DS2 phase. No significant differences in muscle activity were found during the SW phase. For the area under the curve of RMS envelope of EMG, significant differences were observed in the RF, BF, and PL during the DS1 phase; in the MG, TA, and PL during the SS phase; in the BF, MG, TA, and PL during the DS2 phase; and in the BF, TA, and PL during the SW phase. These results showed an asymmetry of muscle activities between left and right legs during walking in healthy subjects.

In the stroke patient group, the differences between the affected and non-affected sides were considered in the left-affected and right-affected sides separately. For the left-affected group, the significant differences in the RMS values were found in the BF and PL during the DS1 phase; in the BF, TA, and PL during the SS phase; in the BF during the DS2 phase; and in the RF and BF during SW phases. For the area under the curve of RMS envelope of EMG, we observed significant differences in all six muscles during the DS1 and SS phases; in the BF, MG, TA, and PL during the DS2 phase; and in the BF, TA and PL during the SW phase.

For the right-affected group, the significant differences in the RMS values were found in the RF, MG, TA and PL during the DS1 phase; in all six muscles during the SS phase; in the GM and TA during the DS2 phase; and in the BF and TA during SW phases. For the area under the curve of RMS envelope of EMG, we observed significant differences in the GM, RF, MG, TA and PL during the DS1 phase; in all six muscles during the SS phases; in the GM, MG, and PL during the DS2 phase; and in the BF and TA during the SW phase.

It suggests that stroke patients have greater asymmetry of lower limb muscle activity in both left-affected and right-affected groups compared to healthy subjects. While the RMS values of the affected side are lower than non-affected side, this difference is particularly evident in the area under the curve of RMS EMG data, with the exception of the RF

and BF muscles of left-affected side group. When comparing the area under the RMS envelope curve to healthy control, stroke patients showed higher values in most muscles across gait subphases. The values were typically highest in the non-affected side, followed by the affected side, and healthy participants.

## Quantitative exploration of muscle activity

The muscle activities of all fifteen healthy participants, for both left and right sides of six muscles during gait cycle, are presented in Figs 3 and 4. The solid lines represent the average muscle activities, while the shaded areas indicate variation. The graphs reveal that the muscle activities exhibited greater variation on the right side compared to the left side in healthy individuals, as evidenced by the wider variation range on the right.

In the healthy group, the GM exhibited dominant activity during the SW phase and continued to function during DS1 through SS in both limbs. The RF shows activity throughout DS1, DS2, and SW phases through less distinctly. The BF displays high activity from DS1 phase through SS, and SW phases, with significant variation throughout the gait cycle. TA has heightened activity during SW, DS1, with a decline during DS2 phase. For MG, the activity of muscle shows a slight increase in activity during DS1, SS, DS2 phases, followed by a decline at the initial of SW phase, with a wide variation. Similarly, the activities of PL increase activity in DS1 phase, peaks during SS phase, and decreases in DS2 phase.

In the stroke patient group (Figs 5 and 6), GM activity increases during the DS1 phase, continues through SS phase, and peaks at the terminal of SW phase on the non-affected side with a wide variation. In contrast, the affected side maintains steady activity throughout the cycle but with a higher variation during the SW phase. The RF shows moderate activity with high variation on both sides, with a higher activity during DS2 phase on the affected side and during SW

**Table 2. Comparison of the RMS value between healthy participants and stroke patients for each affected side group.**

| Subjects | Muscle | Left side | | | | Right side | | | |
|---|---|---|---|---|---|---|---|---|---|
| | | DS1 | SS | DS2 | SW | DS1 | SS | DS2 | SW |
| Healthy | GM | 1.38 (0.07) | 1.42 (0.07) | 1.36 (0.05) | 1.42 (0.08) | 1.36* (0.06) | 1.42 (0.10) | 1.37 (0.06) | 1.43 (0.09) |
| | RF | 1.37 (0.07) | 1.50 (0.13) | 1.36 (0.07) | 1.52 (0.14) | 1.39* (0.07) | 1.50 (0.12) | 1.39* (0.08) | 1.51 (0.12) |
| | BF | 1.45 (0.11) | 1.66 (0.24) | 1.44 (0.12) | 1.56 (0.17) | 1.42 (0.10) | 1.61* (0.19) | 1.43 (0.10) | 1.61 (0.18) |
| | MG | 1.46 (0.13) | 1.64 (0.20) | 1.45 (0.13) | 1.62 (0.21) | 1.46 (0.14) | 1.64 (0.21) | 1.45 (0.12) | 1.64 (0.22) |
| | TA | 1.44 (0.12) | 1.53 (0.15) | 1.41 (0.11) | 1.53 (0.17) | 1.44 (0.12) | 1.52 (0.13) | 1.42 (0.10) | 1.52 (0.16) |
| | PL | 1.39 (0.08) | 1.54 (0.20) | 1.40 (0.12) | 1.53 (0.20) | 1.44* (0.10) | 1.53 (0.15) | 1.42 (0.12) | 1.53 (0.19) |
| Left-affected side group | GM | 1.38 (0.09) | 1.40 (0.08) | 1.37 (0.07) | 1.40 (0.07) | 1.38 (0.07) | 1.40 (0.09) | 1.38 (0.07) | 1.38 (0.07) |
| | RF | 1.50 (0.22) | 1.51 (0.17) | 1.49 (0.18) | 1.51 (0.20) | 1.45 (0.10) | 1.49 (0.11) | 1.43 (0.08) | 1.43* (0.09) |
| | BF | 1.56 (0.23) | 1.58 (0.23) | 1.54 (0.22) | 1.58 (0.21) | 1.40* (0.13) | 1.43* (0.13) | 1.42* (0.14) | 1.41* (0.11) |
| | MG | 1.51 (0.16) | 1.56 (0.19) | 1.51 (0.14) | 1.53 (0.13) | 1.54 (0.13) | 1.60 (0.14) | 1.53 (0.13) | 1.54 (0.10) |
| | TA | 1.47 (0.11) | 1.47 (0.10) | 1.47 (0.11) | 1.51 (0.11) | 1.51 (0.10) | 1.58* (0.15) | 1.50 (0.12) | 1.49 (0.10) |
| | PL | 1.44 (0.10) | 1.51 (0.14) | 1.46 (0.11) | 1.51 (0.14) | 1.52* (0.11) | 1.58 (0.14) | 1.50 (0.10) | 1.55 (0.13) |
| Right-affected side group | GM | 1.37 (0.07) | 1.40 (0.05) | 1.37 (0.04) | 1.38 (0.07) | 1.34 (0.06) | 1.34* (0.05) | 1.34* (0.06) | 1.37 (0.06) |
| | RF | 1.43 (0.11) | 1.48 (0.10) | 1.40 (0.10) | 1.48 (0.13) | 1.38* (0.08) | 1.42* (0.10) | 1.43 (0.12) | 1.45 (0.10) |
| | BF | 1.49 (0.12) | 1.54 (0.08) | 1.48 (0.11) | 1.49 (0.11) | 1.44 (0.11) | 1.46* (0.11) | 1.48 (0.14) | 1.55* (0.13) |
| | MG | 1.56 (0.11) | 1.71 (0.18) | 1.51 (0.14) | 1.56 (0.13) | 1.47* (0.17) | 1.45* (0.09) | 1.45 (0.12) | 1.52 (0.15) |
| | TA | 1.44 (0.08) | 1.52 (0.10) | 1.42 (0.07) | 1.47 (0.10) | 1.35* (0.05) | 1.38* (0.04) | 1.35* (0.04) | 1.40* (0.07) |
| | PL | 1.48 (0.09) | 1.54 (0.12) | 1.46 (0.10) | 1.53 (0.10) | 1.41* (0.18) | 1.41* (0.07) | 1.42 (0.15) | 1.47 (0.12) |

Values are presented as mean (SD).

* $p < 0.05$ Wilcoxon signed rank test compared between limbs in the same group.

**Table 3. Comparison of the area under the curve between RMS envelope of EMG of healthy participants and stroke patients for each affected side group.**

| Subjects | Muscle | Left side | | | | Right side | | | |
|---|---|---|---|---|---|---|---|---|---|
| | | DS1 | SS | DS2 | SW | DS1 | SS | DS2 | SW |
| Healthy | GM | 2.71 (1.85) | 6.40 (3.98) | 2.77 (2.24) | 6.48 (4.13) | 2.42 (1.06) | 5.87 (2.38) | 2.54 (1.51) | 7.01 (5.20) |
| | RF | 2.87 (1.86) | 6.35 (3.50) | 3.14 (2.74) | 6.94 (3.94) | 3.04* (1.88) | 6.33 (2.63) | 2.68 (1.45) | 6.76 (2.31) |
| | BF | 5.26 (3.96) | 12.98 (8.94) | 4.77 (4.81) | 11.53 (8.68) | 8.28* (8.05) | 18.57 (17.66) | 6.46* (5.83) | 18.95* (14.34) |
| | MG | 12.05 (11.81) | 29.29 (18.96) | 11.74 (11.48) | 26.30 (23.15) | 11.15 (10.08) | 22.59* (16.90) | 9.72* (9.58) | 25.63 (24.43) |
| | TA | 11.37 (6.88) | 24.71 (12.53) | 11.30 (6.35) | 26.10 (11.90) | 11.25 (6.85) | 27.60* (15.33) | 13.38* (8.56) | 31.71* (15.80) |
| | PL | 10.40 (6.87) | 32.27 (20.32) | 10.55 (9.50) | 23.11 (16.12) | 15.20* (11.79) | 39.76* (27.67) | 16.00* (11.27) | 31.11* (17.85) |
| Left-affected side group | GM | 3.61 (1.44) | 4.95 (3.33) | 4.36 (2.08) | 6.76 (2.94) | 7.12* (4.90) | 11.69* (10.92) | 5.68 (4.17) | 8.44 (6.28) |
| | RF | 6.34 (3.93) | 9.18 (6.93) | 9.07 (6.03) | 12.80 (7.15) | 12.52* (10.61) | 16.68* (11.26) | 9.78 (7.66) | 12.42 (7.18) |
| | BF | 7.23 (5.44) | 9.85 (7.90) | 8.31 (8.21) | 14.93 (12.89) | 21.44* (21.85) | 33.74* (37.49) | 19.23* (21.89) | 23.70* (22.69) |
| | MG | 10.54 (8.83) | 11.28 (8.13) | 8.99 (5.41) | 14.16 (7.30) | 22.89* (22.09) | 40.12* (40.70) | 15.79* (12.45) | 23.01 (16.33) |
| | TA | 7.32 (5.73) | 11.09 (8.41) | 10.64 (7.47) | 13.22 (7.81) | 21.41* (15.82) | 34.67* (26.68) | 17.73* (11.04) | 21.60* (14.58) |
| | PL | 9.13 (7.34) | 11.62 (10.16) | 9.28 (6.30) | 14.81 (9.23) | 22.14* (14.61) | 39.33* (31.25) | 18.56* (10.37) | 25.37* (11.99) |
| Right-affected side group | GM | 8.09 (5.98) | 12.84 (7.27) | 5.98 (2.50) | 7.88 (5.88) | 4.31* (3.24) | 4.28* (1.72) | 4.53* (2.43) | 7.94 (2.69) |
| | RF | 10.49 (5.90) | 18.07 (6.64) | 9.87 (6.74) | 10.11 (4.72) | 5.83* (2.84) | 8.28* (6.68) | 7.69 (5.32) | 13.74 (6.84) |
| | BF | 10.07 (4.40) | 16.60 (6.19) | 8.51 (3.93) | 11.39 (6.97) | 12.79 (14.43) | 15.78* (24.04) | 13.39 (14.86) | 27.99* (31.51) |
| | MG | 17.14 (14.77) | 33.87 (21.36) | 13.61 (9.26) | 15.38 (11.99) | 5.61* (4.42) | 7.74* (9.37) | 6.43* (6.14) | 10.78 (8.04) |
| | TA | 19.68 (11.05) | 31.06 (13.29) | 14.81 (8.31) | 18.11 (10.28) | 13.18* (7.24) | 17.92* (16.49) | 18.78 (16.08) | 30.25* (20.01) |
| | PL | 20.64 (11.57) | 37.08 (16.46) | 20.00 (12.35) | 20.23 (12.44) | 8.98* (6.17) | 10.46* (8.82) | 8.93* (6.44) | 18.35 (10.21) |

Values are presented as mean (SD).

*p<0.05 Wilcoxon signed rank test compared between limbs in the same group.

phase on the non-affected side. The BF exhibits activity starting from the terminal swing, slightly decreases, and then rises during DS1 phase, remaining stable through SS and DS2 phases with wide variation on the affected side, similar to the non-affected side but with a slightly higher amplitude. The MG muscle shows low amplitude throughout the gait cycle, with a slightly higher amplitude during DS1 phase to initial SS and SW phases. On the non-affected side, it shows higher activity during SW, DS1, SS phases and steep rise during terminal SS phase to DS2 phase with high variation. The TA muscle activity is similar on both sides with a higher amplitude on the non-affected side, except during SW phase where the activity is higher at the terminal phase preparation for the heel strike of the next gait cycle. The PL on the non-affected

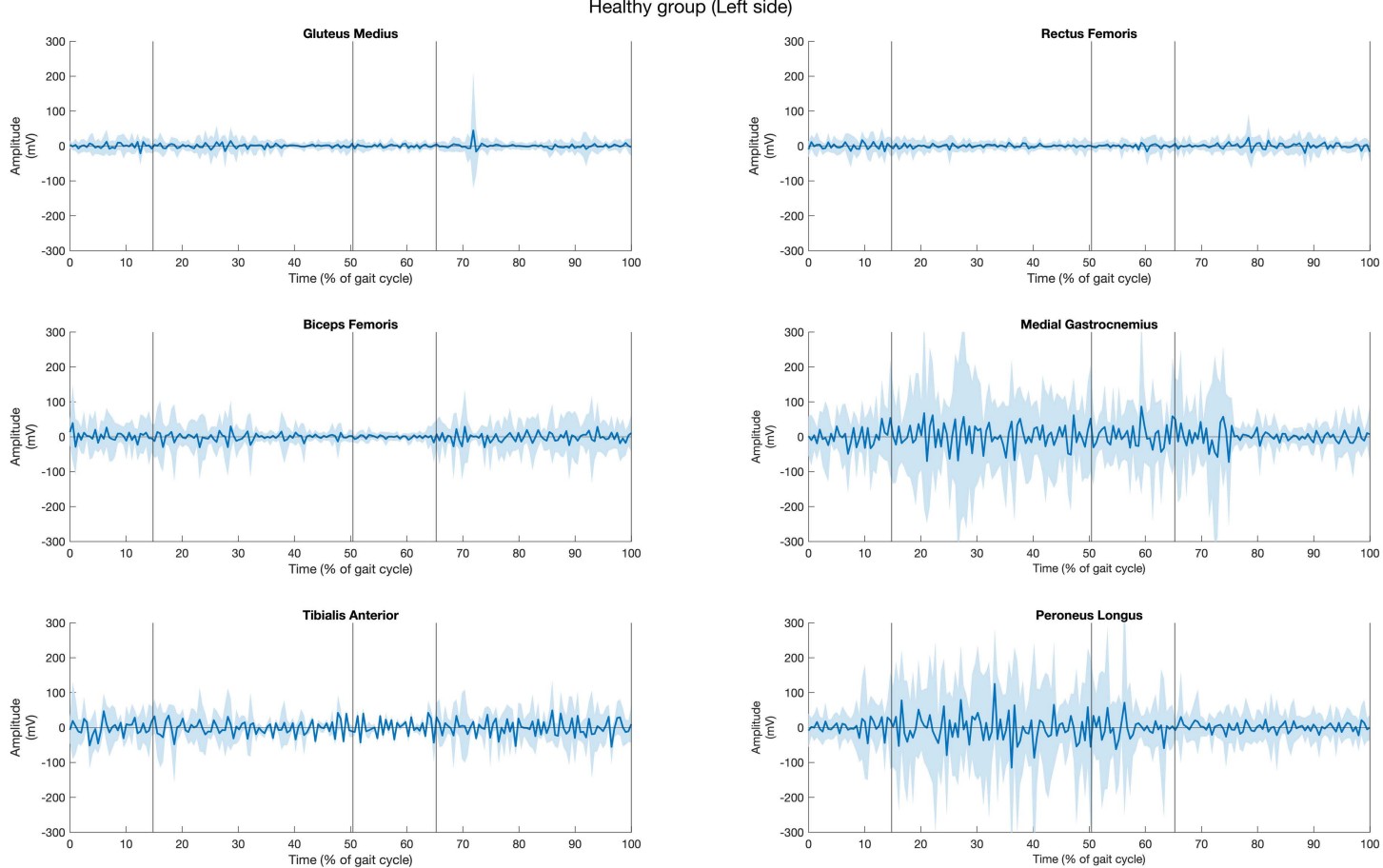

**Fig 3. Muscle activities of gluteus muscle, rectus femoris, biceps femoris, tibialis anterior, medial gastrocnemius, peroneus longus muscle of the left side in the healthy group across the period of (a) first double support (DS1), (b) single support (SS), (c) second double support (DS2), and (d) swing (SW) subphases over one gait cycle. Shaded area represents the range of EMG variability across participants.**

side shows a slight increase in the DS1 phase, followed by a steep rise indicating dominant activity from the SS phase through the DS2 phase, with a high variation, and stable activity during the SW phase.

### Intra-limb correlation

Figs 7 and 8 present significant Spearman $r$ correlations from left and right sides of the healthy group, based on the area under the curve of RMS envelope of EMG. The $r$-values exhibit a similar trend on both sides. Although there are slight differences in the correlation magnitude between left and right sides, the direction of the correlation coefficients remains consistent across all phases of the gait cycle. During the DS1 phase, a significant strong positive correlation was observed between MG and PL ($r_L = 0.74$). In the SS phase, a significant strong positive correlation was found between MG and PL ($r_L = 0.76$), while significant moderate to strong negative correlation was observed between TA and PL ($r_L = -0.68$ and $r_R = -0.74$). In DS2, significant strong positive correlation was found between MG and PL ($r_L = 0.73$). In the SW phase, significant strong positive correlations were found between BF and TA ($r_L = 0.71$), and MG and PL ($r_L = =0.83$).

The results of the correlation analysis for the stroke patients are shown in Figs 9 and 10. Positive correlations were seen in most of the results, in the DS1 phase, there was no significant strong correlation between muscles on either the

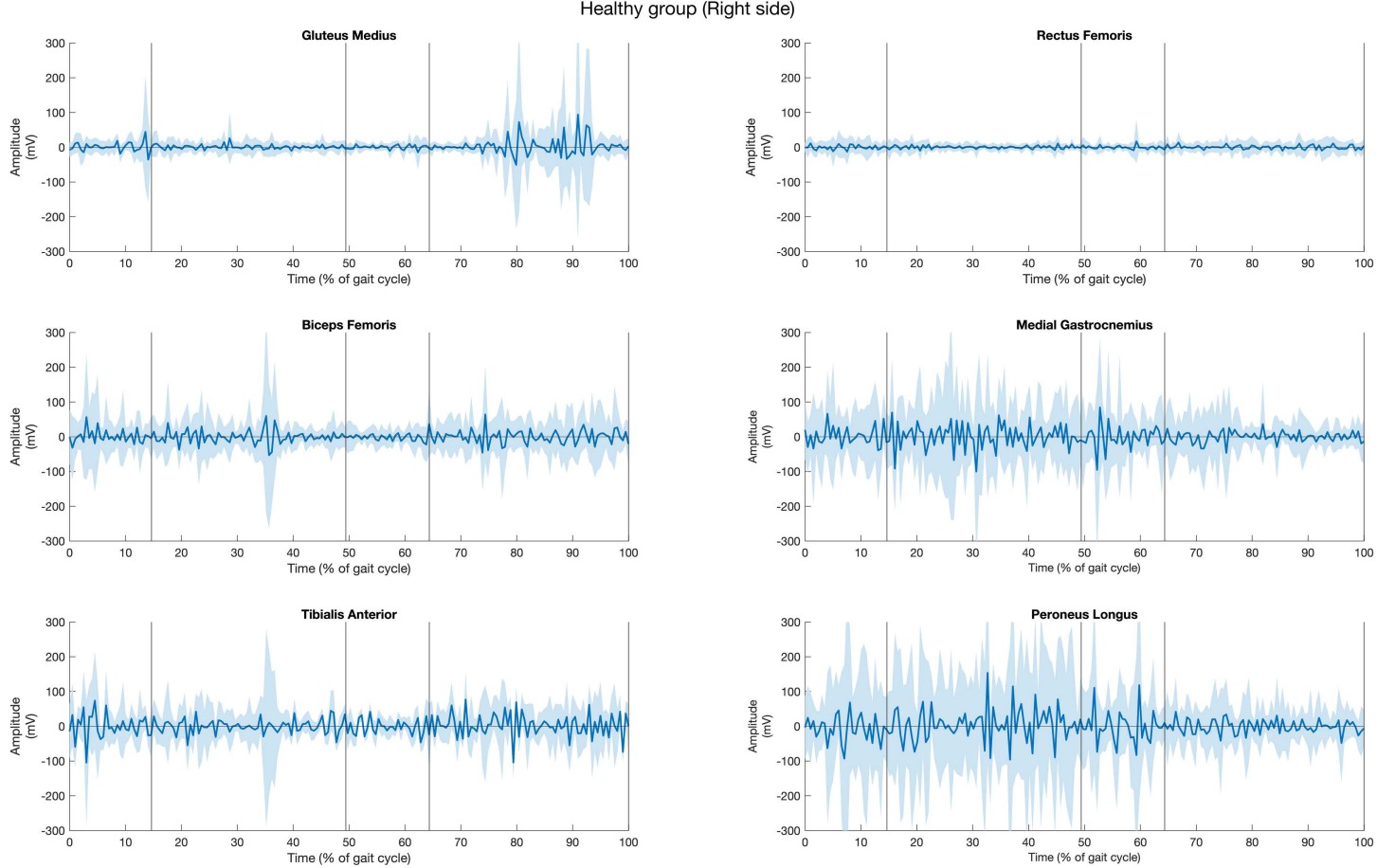

**Fig 4. Muscle activities of gluteus muscle, rectus femoris, biceps femoris, tibialis anterior, medial gastrocnemius, peroneus longus muscle of the right side in healthy group across the period of (a) first double support (DS1), (b) single support (SS), (c) second double support (DS2), and (d) swing (SW) subphases over one gait cycle. Shaded area represents the range of EMG variability across participants.**

affected or non-affected sides. In the SS phase, there was a significant strong positive correlation between GM and RF on the affected side only ($r_{Affected}$=0.75). In the DS2 phase, neither strong positive nor negative correlations were found on the affected and non-affected sides. During the SW phase, no significant strong correlations were found between muscles on both the affected and non-affected sides.

## Inter-limb correlation

The inter-limb correlations for the RMS of EMG envelope in healthy subjects are presented in Fig 11. For the healthy group, during the DS1 phase of left side and the DS2 phase of right side, a strong positive correlation was observed between the MG of left side and the TA of right side during DS1 phase ($r$=0.73). During the SS phase of left side and the SW phase of right side, a strong positive correlation was observed between the TA of left side and the MG of right side ($r$=0.74). In the DS2 phase of left side and the DS1 phase of right side, neither strong positive or negative correlations were found. During the SW phase of left side and the SS phase of right side, strong positive correlations were found between the MG of left side and the TA of right side ($r$=0.81) as well as between the PL of left side and the TA of right side ($r$=0.73), while a strong negative correlation was observed between the MG of left side and the PL of right side ($r$=-0.71).

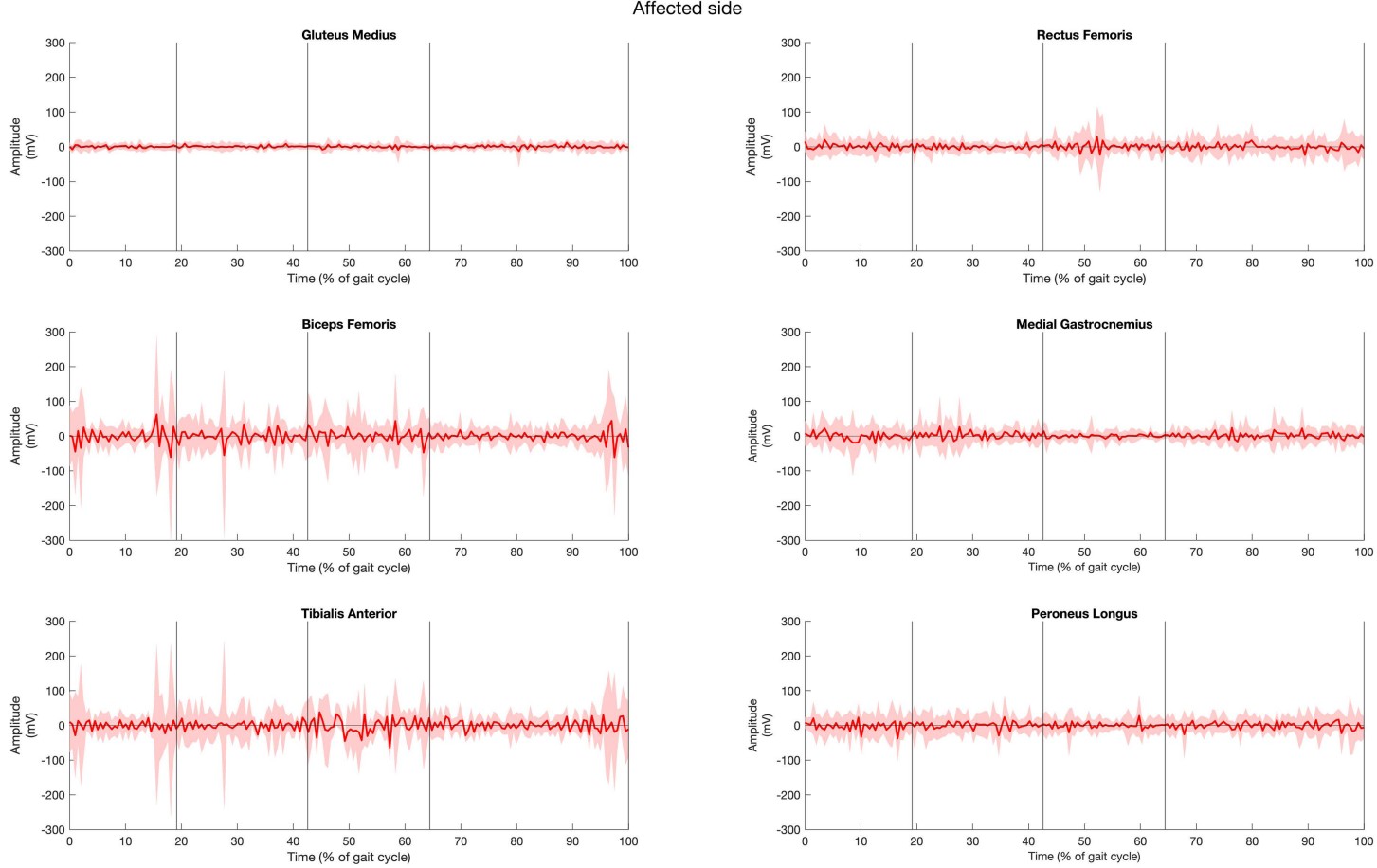

**Fig 5. Muscle activities of gluteus muscle, rectus femoris, biceps femoris, tibialis anterior, medial gastrocnemius, peroneus longus muscle of the affected side in stroke group across the period of (a) first double support (DS1), (b) single support (SS), (c) second double support (DS2), and (d) swing (SW) subphases over one gait cycle.**

For the stroke patient group, the inter-limb correlations of muscles for affected and non-affected limbs are presented in Fig 12. In all phases, neither strong positive nor negative correlations were found. Only moderate positive correlations were detectable such as between the GM and RF of affected side during the SS phase and the GM of non-affected during the SW phase It is clearly showed that healthy subjects have more negative inter-limb correlations than stroke patients.

## Discussion

The present study was designed to determine the correlation among lower limb muscle activities during walking in stroke patients compared to healthy individuals. The findings indicate notable disparities in muscle activation across all six muscles in both healthy subjects and stroke patients. The analysis of RMS values and the area under the curve of RMS envelope of EMG reveals asymmetry in muscle activation during stance phase between left and right sides, with more balanced muscle activation during the swing phase in healthy participants. This finding was supported by our previous study which showed asymmetry in plantar pressure between both feet in healthy subjects [19]. The results highlight differences in muscle activaion and the level of effort exerted by muscles throughout the various phases of gait cycle. In the stroke patient either left-affected or right-affected groups, the outcomes align with previous research highlighting

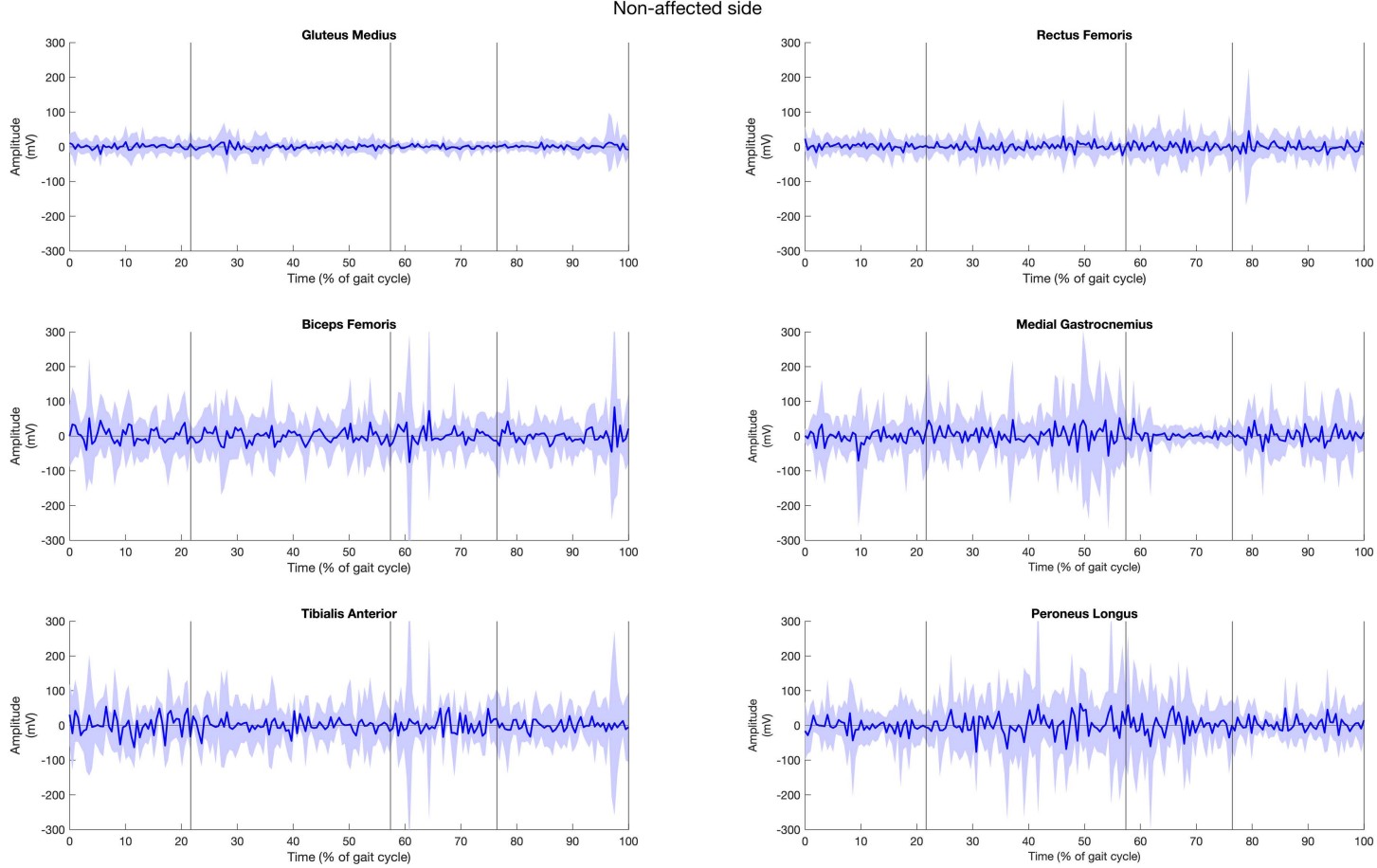

**Fig 6. Muscle activities of gluteus muscle, rectus femoris, biceps femoris, tibialis anterior, medial gastrocnemius, peroneus longus muscle of the non-affected side in stroke group across the period of (a) first double support (DS1), (b) single support (SS), (c) second double support (DS2), and (d) swing (SW) subphases over one gait cycle.**

more asymmetry between limbs which will potentially affect stability [3]. The RMS values of affected side were generally lower than non-affected side in both groups, except for the RF and BF muscles of left-affected group. The increased RMS values of these muscles may indicate spasticity, potentially contributing to the abnormality of knee movement patterns in this group [3]. According to prior studies [11,20], poorer gait performance is associated with slower walking speeds, which aligns with our findings that the left-affected group exhibited slower walking speeds compared to right-affected group. This abnormality in knee movement patterns could be a factor contributing to the poorer performance observed in the left-affected group. The higher area under RMS envelope curve values of stroke patients compared to healthy individuals possibly due to stroke disrupting normal movement coordination, which alters the timing and intensity of muscle activation in stroke patients [3,9]. The non-affected side may compensate to ensure safer movement, for example, by increasing the activity of the GM which plays a key role in excessive hip abduction commonly observed in stroke patients with a stiff knee gait. This compensation resulting in a motion known as circumduction, which is not typically associated with the primary function of GM in hip flexion, leading to higher activity in stroke patients compared to healthy individuals [21].

A gait cycle begins with the heel strike and ends with the subsequent heel strike of the ipsilateral side. In DS1 stage, the movement of the lower limb involves hip flexion, knee extension, and ankle dorsiflexion to a neutral position

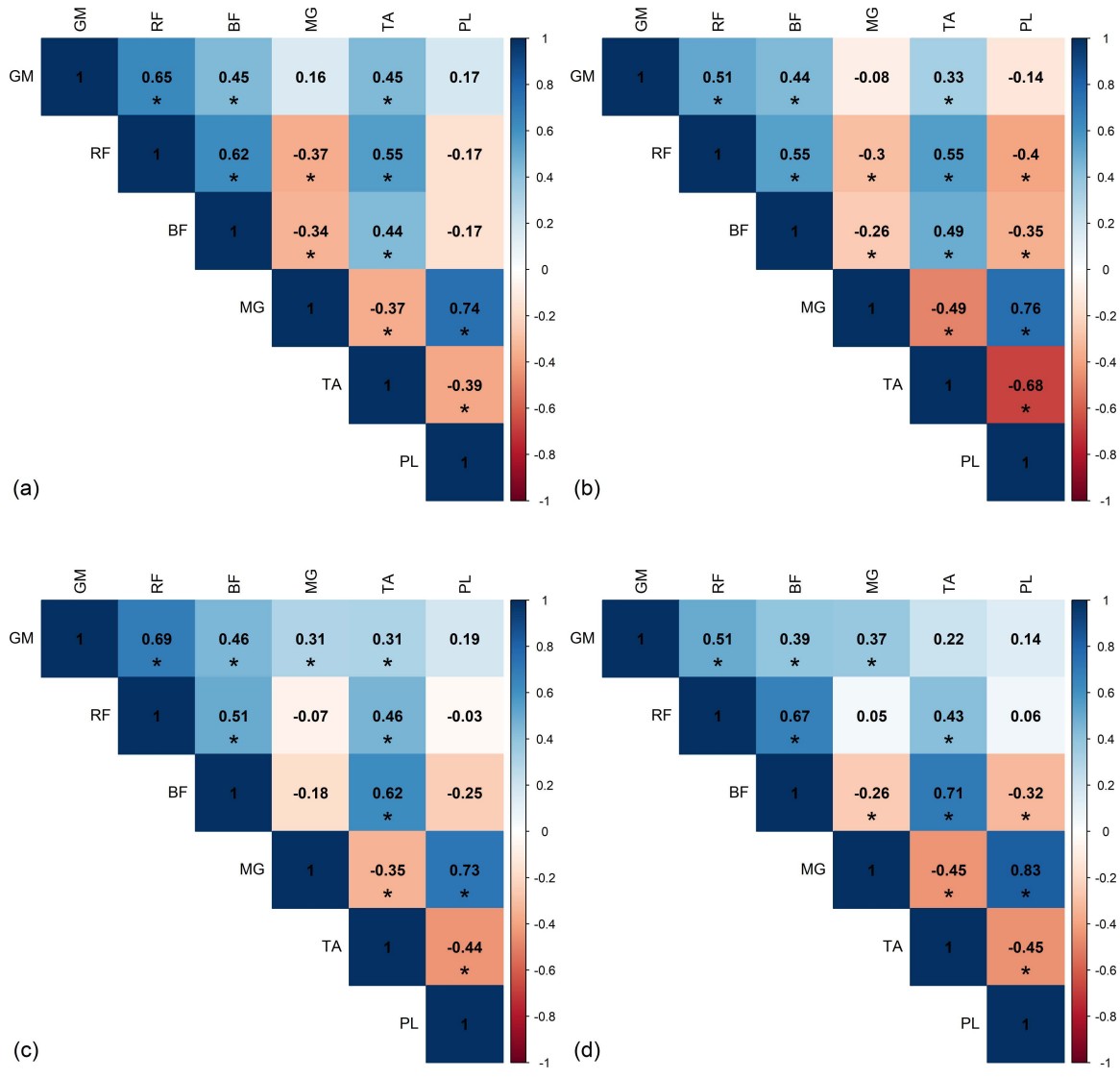

**Fig 7. The significant intra-limb Spearman's *r*-value from left side of healthy group in (a) first double support phase (DS1), (b) single support phase (SS), (c) second double support phase (DS2), and (d) swing phase (SW). GM, gluteus medius muscle; RF, rectus femoris muscle; BF, biceps femoris muscle; MG, medial gastrocnemius muscle; TA, tibialis anterior muscle; PL, peroneus longus muscle.**

[10]. The present findings indicated that the RF, one of the quadricepts muscles located at the anterior thigh, assists in enhancing movement at the hip and knee joints. The RF contracts to flex the hip while simultaneously extending the knee [22]. Additionally, the BF, a hamstring muscle located at the posterior thigh, co-contracts with the RF to produce smooth hip joint motion, which explains the observed RF activity during the DS1 stage [23]. In addition, the TA is responsible for ankle dorsiflexion, while both the MG and PL assist in regulating the rate of plantarflexion and stabilizing the foot and ankle during this period. Although the GM is part of the hip extensor group, it does not contribute to hip flexion [24], therefore, the observed correlation between its activity on both sides was not align with our findings, as it is not the primary muscle involved in the movement during this phase.

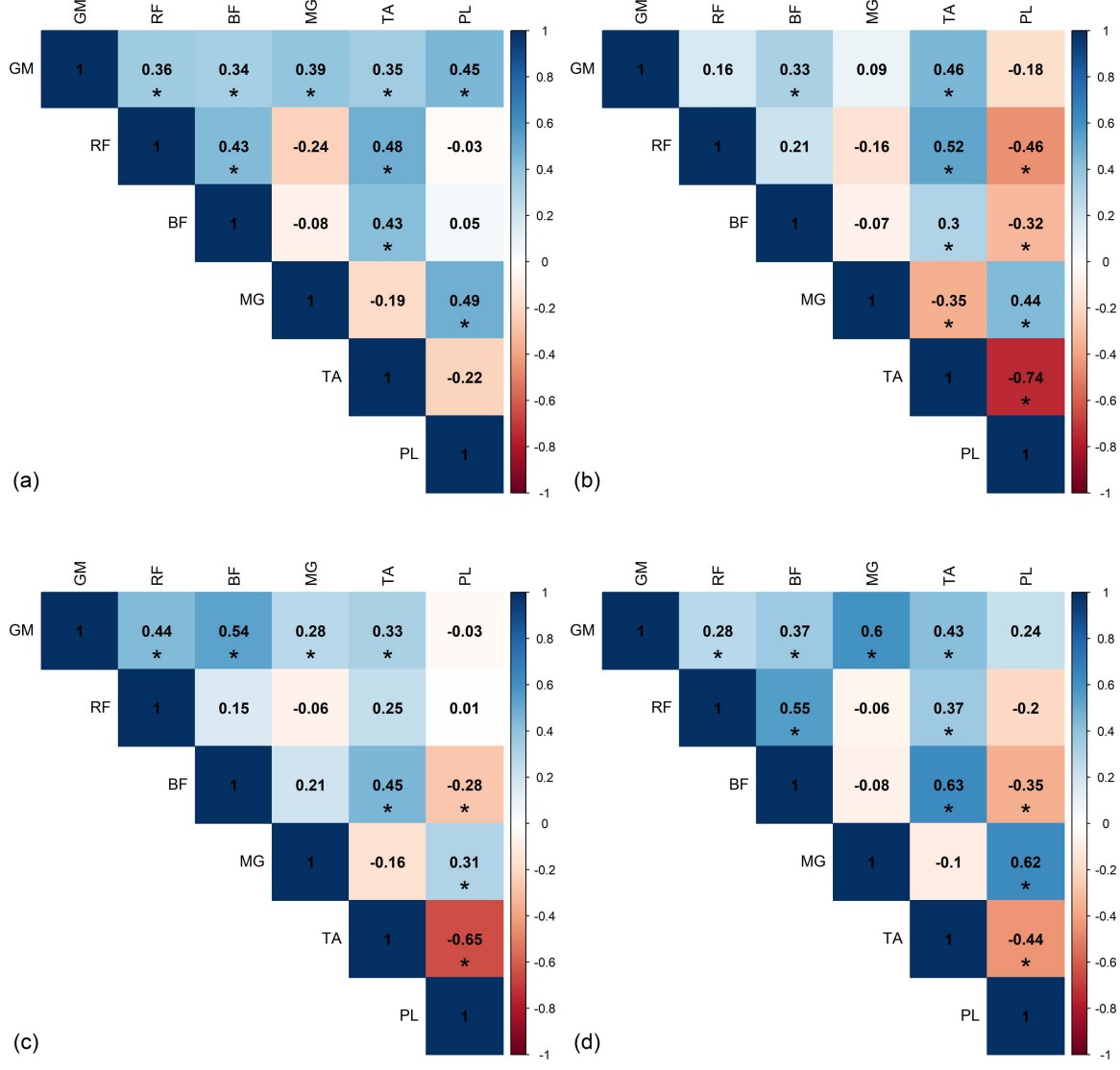

**Fig 8. The significant intra-limb Spearman's *r*-value from right side of healthy group in (a) first double support phase (DS1), (b) single support phase (SS), (c) second double support phase (DS2), and (d) swing phase (SW).** GM, gluteus medius muscle; RF, rectus femoris muscle; BF, biceps femoris muscle; MG, medial gastrocnemius muscle; TA, tibialis anterior muscle; PL, peroneus longus muscle.

The SS phase, as illustrated in Fig 1, begins with contralateral toe off to initiate swing and continues until the contralateral foot contacts the ground [10,15]. During this stage, the movement of the lower limb involves hip and knee extension, as well as ankle dorsiflexion [10]. To ensure safe and effective movement throughout this phase relies on the activity of knee extensors, with the RF playing an important role in controlling knee extension and stabilizing the knee joint [23], as observed in this study. The intense activity of hip and knee muscles during loading response diminishes by early mid-stance, transitioning to a more stabilized muscle engagement as the body moves forward. Limb stability supported by the MG [10]. In term of ankle joint movement, the TA, MG, and PL are primary contributors and we observed that the dominant activities of MG and PL exhibit an inverse relationship to TA activity, consistent with Kumar et al. [23]. The TA

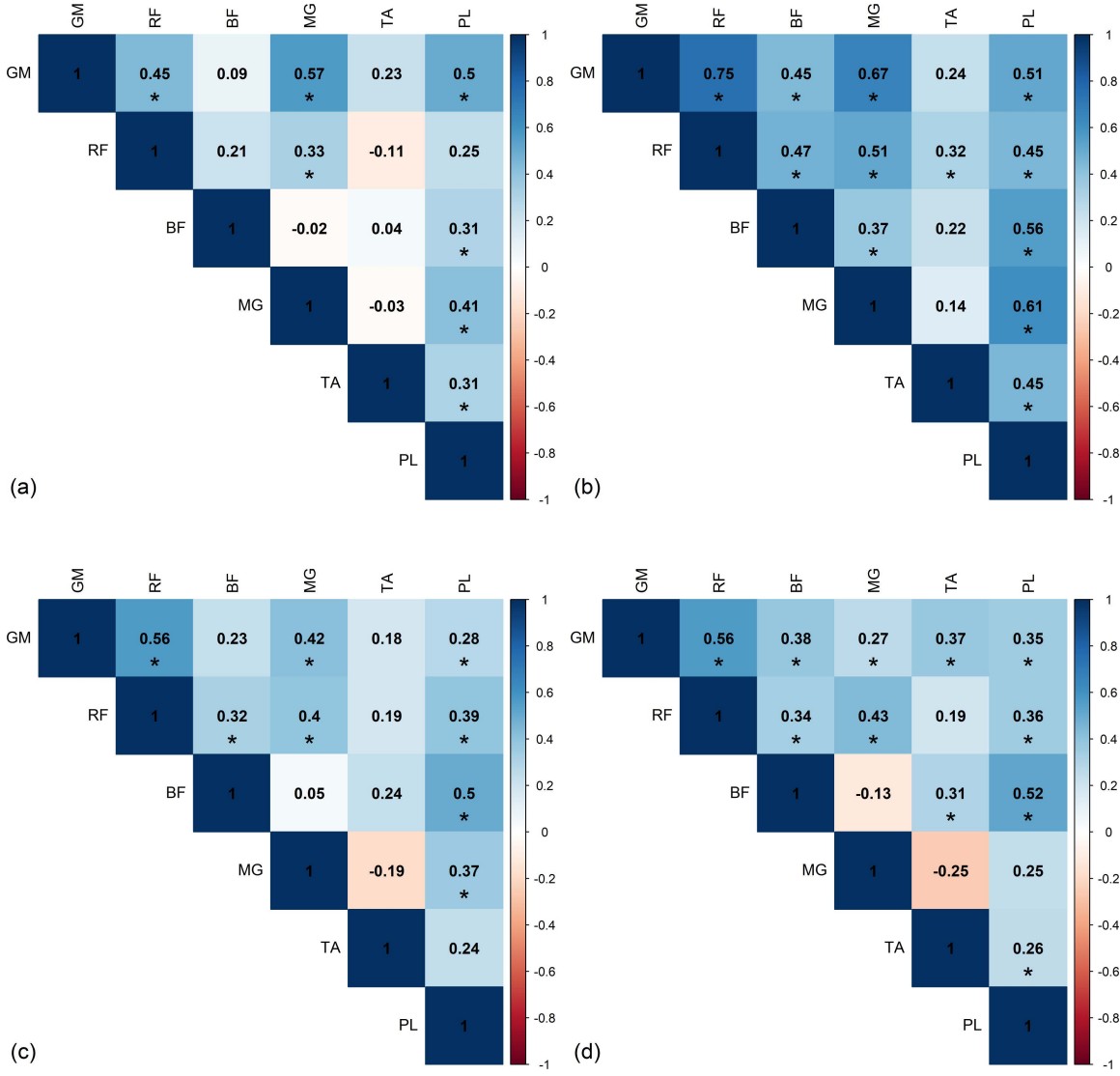

**Fig 9. The significant intra-limb Spearman's *r*-value from affected side of stroke group in (a) first double support phase (DS1), (b) single support phase (SS), (c) second double support phase (DS2), and (d) swing phase (SW).** GM, gluteus medius muscle; RF, rectus femoris muscle; BF, biceps femoris muscle; MG, medial gastrocnemius muscle; TA, tibialis anterior muscle; PL, peroneus longus muscle.

is responsible for ankle dorsiflexion, while the MG and PL help regulate plantar flexion during walking and these muscles also play a major role in maintaining foot posture [23].

The DS2 phase commences as the contralateral foot contacts the ground during heel strike and continues until the ipsilateral foot initiates its swing phase at toe-off. The ipsilateral limb exhibits reduced hip extension, increased knee flexion, and ankle plantar flexion in response to the initial weight transfer while the contralateral limb is loaded [10]. In this phase, the lower limb undergoes external rotation to maintain stability, followed by internal rotation in preparation for toe clearance during initial swing [10]. The GM, primarily involved in hip abduction, also functions as a secondary internal hip rotator, crucial for stabilizing the hip and assisting the BF in controlling thigh movement for hip extension during stance. The RF initiates knee extension, while the BF continues the motion through knee flexion. Regarding the ankle joint, the

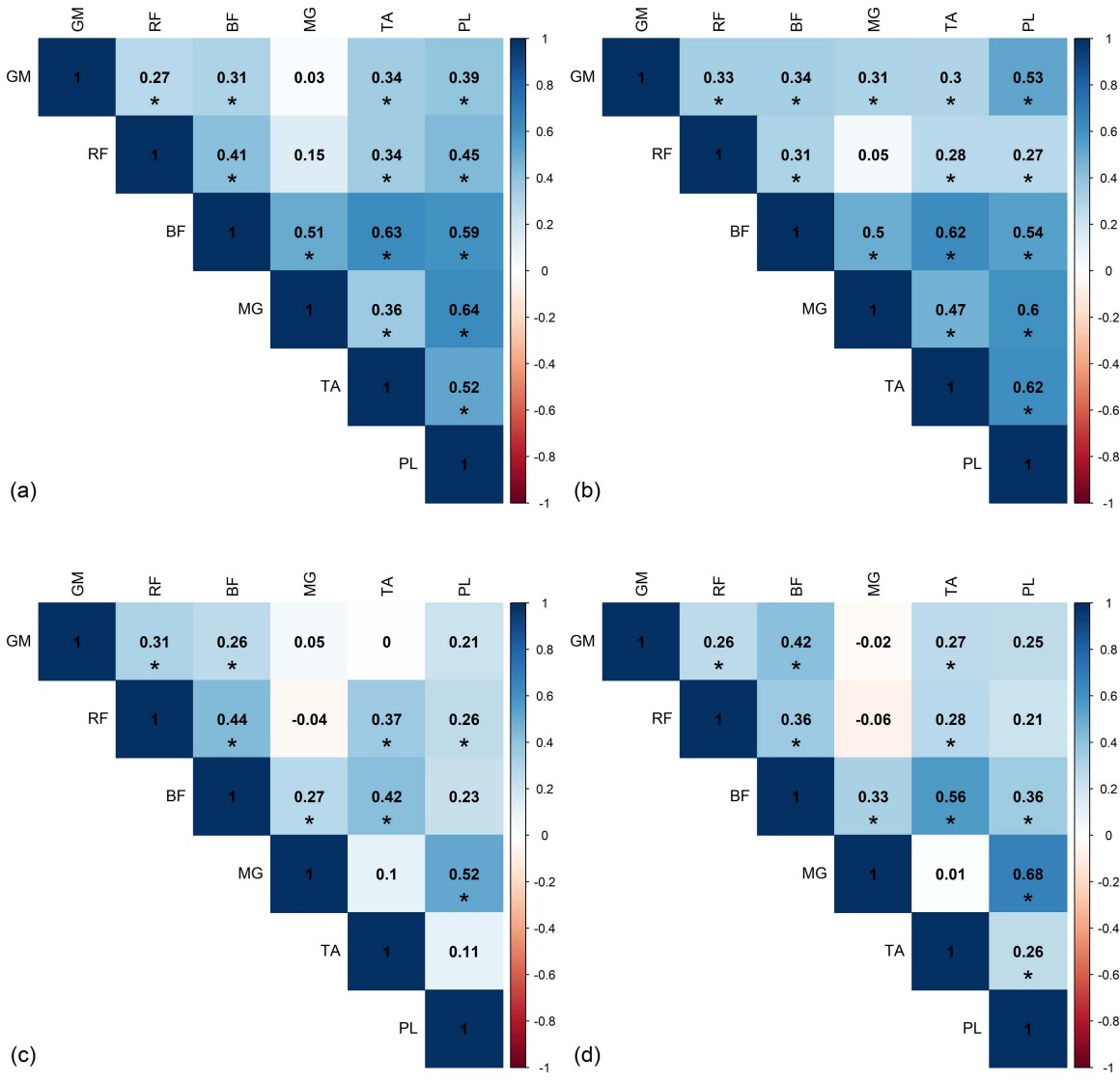

**Fig 10. The significant intra-limb Spearman's *r*-value from non-affected side of stroke group in (a) first double support phase (DS1), (b) single support phase (SS), (c) second double support phase (DS2), and (d) swing phase (SW). GM, gluteus medius muscle; RF, rectus femoris muscle; BF, biceps femoris muscle; MG, medial gastrocnemius muscle; TA, tibialis anterior muscle; PL, peroneus longus muscle.**

MG and PL have roles in maintaining balance and stability, as well as ensuring efficient weight transfer. Both muscles are primary facilitators of plantar flexion. Their coordinate activity is vital for proper locomotion and foot posture.

In the final period of the gait cycle, the SW phase begins when the ipsilateral foot lifts from the floor and ends when the same foot strikes the floor. This phase, known as the limb advancement period, encompasses weight release, forward limb progression, clearance of the ipsilateral foot from the floor, and preparation for the subsequent gait cycle [10]. Limb progression is achieved through hip flexion and ankle dorsiflexion to neutral position. During this period, the knee transitions from flexion to full extension. In this study, the BF was observed to play a key role in facilitating knee flexion. Concurrently, ankle dorsiflexion is supported by the TA which is the most powerful dorsiflexor muscle of the foot, working in coordination with the MG and PL, which assist in plantar flexion.

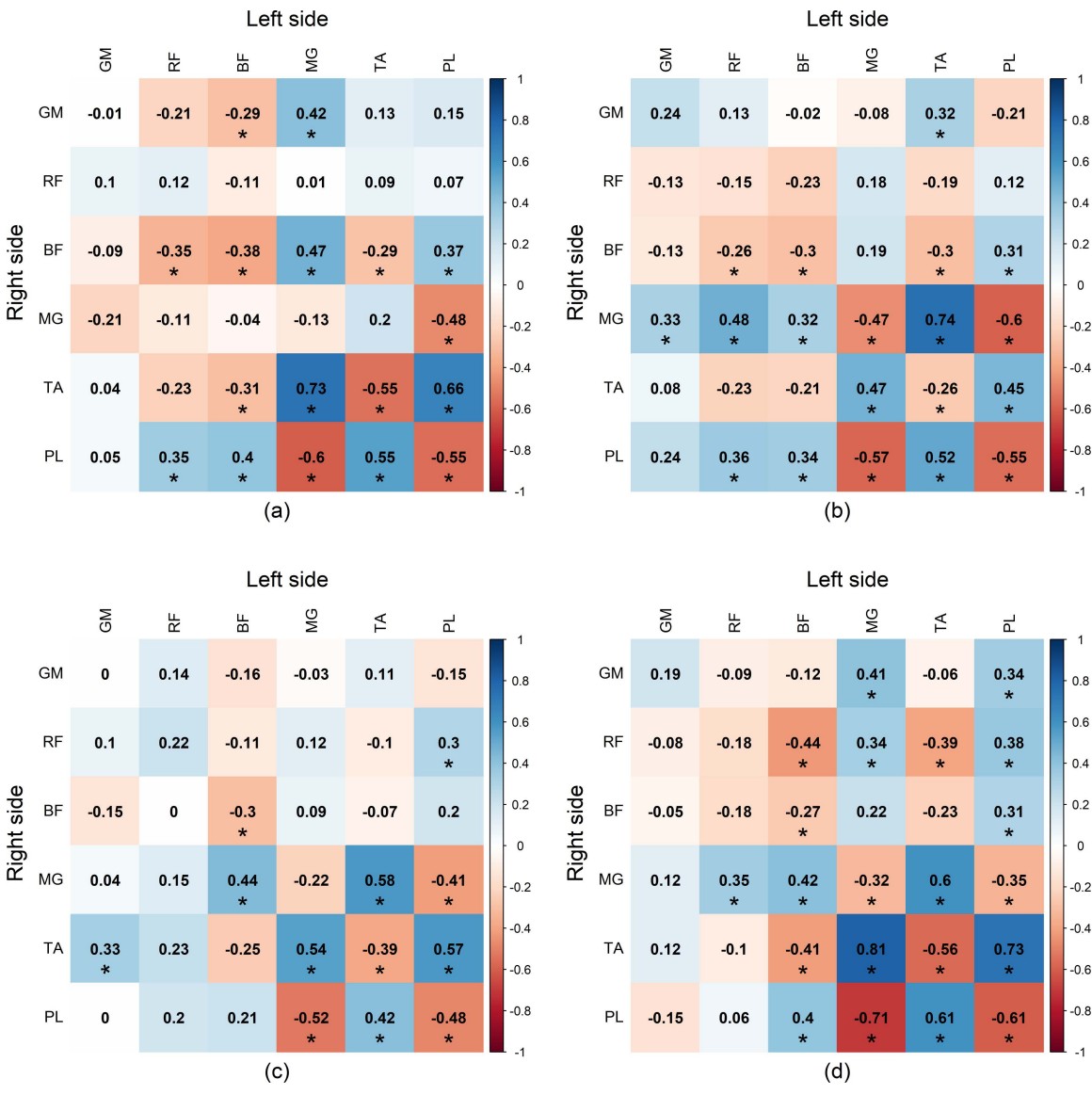

**Fig 11. Significant inter-limb Spearman's *r*-value between the left and right sides in healthy group during (a) first double support phase (DS1), (b) single support phase (SS), (c) second double support phase (DS2), and (d) swing phase (SW).** GM, gluteus medius muscle; RF, rectus femoris muscle; BF, biceps femoris muscle; MG, medial gastrocnemius muscle; TA, tibialis anterior muscle; PL, peroneus longus muscle.

In stroke patients compared to healthy participants, most striking was that the only strong correlation was between the GM and RF activity during single support of the affected side, during which the GM plays a dominant role in stabilizing the pelvis and maintaining balance, while the RF primarily assists in hip flexion and knee extension. Commonly seen gait impairments in stroke patients include knee hyperextension during stance and stiff knee gait during swing phases [3]. Knee hyperextension in stroke patients can be caused by decreased strength of the hip extensors, including the GM, and passive stability of the knee, thereby increasing the demand on the quadriceps muscle, particularly the RF, or by weakness in the ankle plantar flexors, particularly the MG [25,26]. Stiff knee gait, which can result from several factors such as excessive of the RF activity, insufficient ankle push-off and reduced strength in the hip extensor and ankle plantar flexor

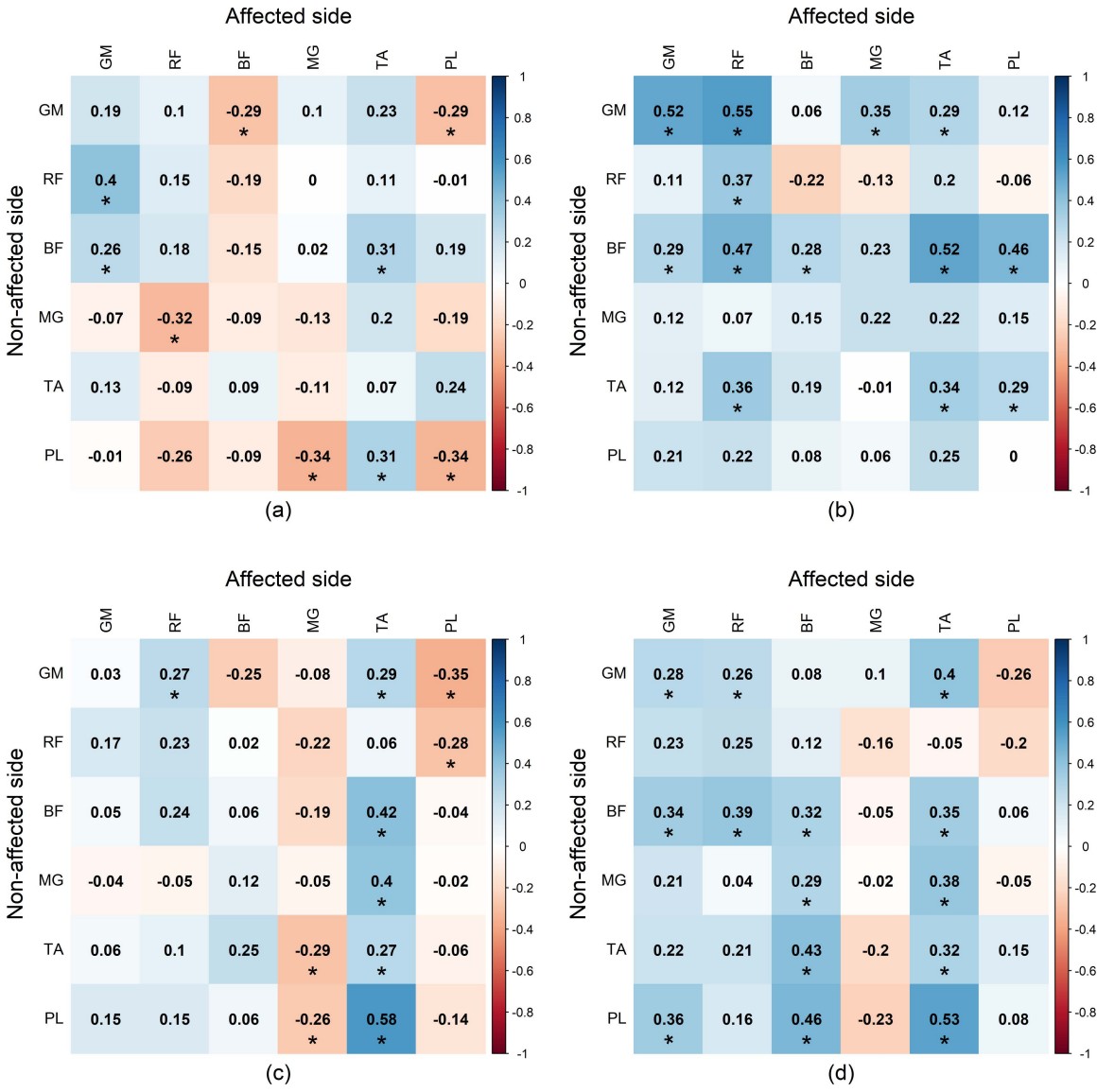

**Fig 12. Significant inter-limb Spearman's *r*-value between the affected and non-affected sides in stroke group during (a) first double support phase (DS1), (b) single support phase (SS), (c) second double support phase (DS2), and (d) swing phase (SW).** GM, gluteus medius muscle; RF, rectus femoris muscle; BF, biceps femoris muscle; MG, medial gastrocnemius muscle; TA, tibialis anterior muscle; PL, peroneus longus muscle.

muscles [27]. This study similar to the work of Akbas et al. [21], but with a notable observation of dominant activity during the single support phase. Our findings reveal that the RF-MG coupling in post-stroke gait is related to abnormal coordination patterns. Such patterns include vaulting and pelvic obliquity, which may compensate for a lack of knee flexion, and circumduction, which serves as compensation in stiff knee gait [21]. Foot drop is another common disturbance during the swing phase, caused by insufficient function of the TA due to impaired dorsiflexion, or from spasticity of the plantar flexion muscle [3,9]. Altered gait patterns in stroke patients, such as stiff knee gait, can reduce foot clearance contributing to foot drop during the swing phase [3].

Focusing the inter-limb correlation, in the healthy group, the notable finding was the presence of moderate to strong correlations predominantly in the distal leg muscles including the MG, TA, and PL throughout all phases of the gait cycle. This could be due to various factors, such as declines in sensory and motor processes. Impairments in motor systems often involve both neuromuscular and musculoskeletal systems [28]. Gait characteristics typically begin to show mild transitional changes with aging [29]. Normal aging is characterized by changes in gait pattern, such as decreased gait velocity, reduced stride length, diminished peak knee flexion during the swing phase, and slightly increased ankle dorsi-flexion [29]. In middle adulthood (40s and 50s), muscle contraction slows with a 30% decline in muscle strength from age 50–70, followed by a more rapid decline after age 70 [29]. Muscle fiber loss in both type I and type II fibers is associated with aging. It has been noted that type II muscles are lost more quickly than type I fibers, aligning with the observed age-related reduction in the number of motor units and the size of type II muscle fibers, particularly in the proximal muscles [28–30]. The intra-limb findings of our study align with those observed in the earlier studies, indicating that the dominant activity in healthy participants is associated with weakness in the proximal lower limb muscles. This weakness heightens the risk of falls in older adults following muscle fatigue, as proximal muscle weakness leads to greater reliance on distal muscles during high-intensity activities [31].

In contrast to stroke participants, a predominantly moderate positive correlation was observed in the proximal muscles across all phases of the gait cycle. This may attributed to greater impairment of distal muscles compared to proximal lower limb muscles, consistent with the findings of Andrew and Bohannon [32] who observed that proximal muscles less impairment than distal muscles. Lin et al. suggested that the muscle weakness or fatigue at the ankle has a greater impact than fatigue at the knee [33]. Likewise, Horling et al. found that individuals with distal muscle weakness experienced greater instability during forward and backward perturbations, whereas those with proximal muscle weakness were only unstable during backward perturbations [34]. In our study, intense activity of the GM, BF, and TA was observed on the affected side during the DS1, DS2, and SW phases, likely due to weakness in the knee and ankle. During the SS phase, activity of the GM, RF, TA, and PL was noted, possibly to prevent excessive hip drop during contralateral swing, facilitate high activity of RF due to knee hyperextension, and stabilize the ankle to support body weight [10,23]. Reduced activity in the BF was observed in stroke patients, potentially linked to their inability to achieve effective knee flexion, which aligns with previous findings associating this impairment with altered muscle activation patterns and simplified synergies during gait [35]. On the non-affected side, altered activity was observed in all six muscles. Our findings are consistent with Chow et al., [36] which observed a co-contraction of the TA, MG, and PL, typically antagonist muscles that ensure smooth ankle motion.

Our study suggests a potential training approach based on age-related weakness in proximal muscles among healthy individuals, which could heighten the risk of falls when distal muscles are intensively used, leading to increased local fatigue. Therefore, prioritizing training for proximal lower limb muscles may enhance muscle endurance and strength. For stroke patients, particularly elderly individuals, if weakness predominantly affects distal muscles, physiotherapy should be concentrated not only on strengthening the distal muscles of the affected limb but also on training the core body and proximal leg muscles to improve symmetry and support for weight-bearing. However, this current study was constrained by the lack of information regarding the specific locations of brain lesions in stroke patients and the dominant side in both groups, which importantly impacts gait performance. The specific location and extent of the brain injury sustained by stroke patients can significantly affect their motor control and coordination, leading to diverse gait impairment. Additionally, the study was limited by a relatively small sample size, particularly among stroke patients. A larger sample size is needed to better understand the factors related to gait function in post-stroke patients. Analyzing the impact of specific brain lesions on gait performance and including a more diverse and larger patient population would provide a more comprehensive understanding of the factors influencing gait function in post-stroke individuals. Furthermore, the variability in EMG measurement (e.g., electrode placement and skin preparation) and stroke severity among participants may influence the results. Incorporating the Vicon systems for kinematics validation has also been highlighted as a potential approach to enhance the scientific rigor of the study. It is necessary to consider these influences in future studies.

## Conclusion

This study investigated the correlation of lower limb muscle activities during walking in stroke patients compared to healthy individuals, highlighting significant disparities in muscle activation. The findings indicate pronounced differences in activation during the stance phase for healthy individuals and across the entire gait cycle for stroke patients. Key finding in the stroke group highlights that the RF and GM muscles are essential to gait stability, particularly during single support. Stroke patients exhibited knee hyperextension during stance often due to muscle weakness and compensatory strategies, contributing to abnormal coordination patterns and potentially coexisting with stiff knee gait during swing phase. In healthy individuals, minor asymmetry might not heavily impact walking, but it is important to monitor and correct any emerging imbalances to prevent potential issues, such as an increased risk of falls. For stroke patients, the observed asymmetry in muscle activation throughout all gait phases indicates impaired motor control and muscle coordination resulting from neurological damage. These insights are valuable for rehabilitation professionals, such as physiotherapists, to design rehabilitation programs for stroke patients. Rehabilitation efforts should focus on reducing muscle activity asymmetries through targeted exercises and therapies to improve balance, strength, and overall gait stability.

## Acknowledgments

Our gratitude extends to Assistant Professor Chatwalai Sonthikul for valuable insight on addressing the clinical implications of stroke. We also thank all participants for their contributions to this study.

## Author contributions

**Conceptualization:** Thanita Sanghan, Surapong Chatpun.

**Data curation:** Thanita Sanghan, Nusreena Hohsoh.

**Formal analysis:** Thanita Sanghan, Surapong Chatpun.

**Funding acquisition:** Goran Stojanović, Surapong Chatpun.

**Investigation:** Thanita Sanghan, Nusreena Hohsoh.

**Methodology:** Thanita Sanghan, Surapong Chatpun.

**Supervision:** Goran Stojanović, Rezaul Begg, Surapong Chatpun.

**Visualization:** Thanita Sanghan.

**Writing – original draft:** Thanita Sanghan, Nusreena Hohsoh.

**Writing – review & editing:** Rezaul Begg, Surapong Chatpun.

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
