## [Decision Letter · Decision Letter 0]

30 Oct 2024

PONE-D-24-38957Lower limb muscle activity during walking: A comparative study of stroke patients and healthy individualsPLOS ONE

Dear Dr. Chatpun,

Thank you for submitting your manuscript to PLOS ONE. After careful consideration, we feel that it has merit but does not fully meet PLOS ONE’s publication criteria as it currently stands. Therefore, we invite you to submit a revised version of the manuscript that addresses the points raised during the review process.

We look forward to receiving your revised manuscript.

Kind regards,

Tomoyoshi Komiyama, Ph.D

Academic Editor

PLOS ONE

Journal Requirements:

Additional Editor Comments:

Dear authors,

Your research investigated the correlation between lower limb muscle activity during walking in stroke patients and healthy subjects, and found significant differences. Your results also showed significant differences in muscle activity during the stance phase in healthy subjects and throughout the gait cycle in stroke patients.

However, I think that it is necessary to strengthen the reliability of these results by adding as much information as possible.

We thus have some questions and suggestions for the manuscript that you might consider.

I believe these comments will be helpful in the revision of your study.

Also, as per my comments, I recommend that the conclusion section is made clearer.

The purpose of this study should also be made clear through explanations in the discussion section.

For example, who can it help and how?

Tomoyoshi Komiyama

Reviewers' comments:

Reviewer's Responses to Questions

**Comments to the Author**

1. Is the manuscript technically sound, and do the data support the conclusions?

Reviewer #1: Yes

Reviewer #2: Partly

2. Has the statistical analysis been performed appropriately and rigorously? 

Reviewer #1: Yes

Reviewer #2: Yes

3. Have the authors made all data underlying the findings in their manuscript fully available?

Reviewer #1: No

Reviewer #2: Yes

4. Is the manuscript presented in an intelligible fashion and written in standard English?

Reviewer #1: Yes

Reviewer #2: No

5. Review Comments to the Author

Reviewer #1: The study investigates gait analysis in stroke patients compared to healthy individuals, which is crucial for understanding post-stroke rehabilitation challenges. The selection of muscles and gait phases (DS1, SS, DS2, SW) seems appropriate and the findings could have significant applications. Comparing stroke survivors to healthy subjects adds valuable context, especially the insights derived from the healthy cohort, which are particularly new and interesting.

The manuscript presents its concepts clearly and concisely. The overall quality of writing in commendable, making the paper easy to follow.

However, some issues need attention:

- Line 22: The paper lists muscles but omits the tibialis anterior (TA).

- Several references cited in the discussion are not introduced in the introduction, which disrupts the flow of the manuscript and the background context.

- It is not specified how the EMG sensors were placed. Did you follow some guidelines? Providing this information would enhance the repeatability of the study.

- The type and order of filters used to process EMG signals are not detailed. Additionally, there is no explanation provided for how the envelope was computed.

- The software used for the EMG analysis is not mentioned, which is important for replicating the analysis.

- Using only normative percentages to extract gait sub-cycles might not be the most accurate approach, especially given the variability among stroke patients. It might be more appropriate to use Vicon markers to manually segment the gait cycle, which could better account for individual differences in gait, particularly in stroke patients.

- As a potential future direction, as you are using Vicon you could also collect kinematic data (e.g., joint angles). This would allow you to validate claims, such as knee hyperextension, with concrete kinematic evidence.

Regarding figures and tables:

- It would be better to place figure captions also with each figure, as it can be frustrating for readers to flip back and forth between the text and figures.

- In Figure 1, it would be helpful to include abbreviation for the gait cycles in the caption for clarity.

- The ellipsis (“…”) before (1) in Equation (1) should be removed to maintain proper formatting.

- From my point of view, the distinction between the meaning of RMS and RMS of the envelope is not sufficiently explained. A deeper explanation is needed to clarify the purpose of each and why one is used over the other in specific contexts.

- The quality of the figures in the paper is suboptimal. Although they improve when downloaded, it would be beneficial to enhance their resolution in the manuscript itself, so that downloading is not strictly necessary.

- The presentation of Tables 2 and 3 is confusing. You place the asterisk (*) on the right side only, which gives the impression that where are differences only on the right side, although here you are comparing both right and left. Including p-values for the comparison and marking there the significance difference between both sides would make it clearer that you are comparing between limbs.

- There is also a lack of explanation for why some differences that exist in healthy subjects are absent in stroke patients, as shown when comparing Table 2 and 3. Maybe this should be discussed.

- For figures 2 and 3, you should add a legend or a caption clarifying that the shaded area represents all cycles and the line represents the mean, as this isn’t immediately obvious.

- In figure 2 replace “normal person” with “healthy subject” or “healthy control” for a greater consistency and accuracy.

- The footnotes for figure 6 and 7 are missing.

- The orientation of the figures should be standardized (horizontal or vertical).

- In the discussion, an insufficient ankle push-off and reduced ankle plantar flexor strength were mentioned. Why was not this linked to foot drop phenomenon, a common characteristic in stroke patients?

The paper mentions limitations like small sample size, but there are other areas that could benefit from further discussion:

-Variability in EMG measurements, especially given the potential differences in sensor placement and skin preparation.

- Differences in stroke severity among participants, which could influence the results.

Conclusion

The study presents interesting and clinically relevant insights into muscle activity during walking in stroke patients compared to healthy individuals. However, several methodological details, figure presentations and analytical explanations need to be clarified or improved to strengthen the manuscript. Addressing these issues will enhance the rigor and clarity of the study, making it more impactful for both clinical and academic audiences.

Reviewer #2: The authors undertook the difficult task of conducting experimental research with stroke patients - physically and emotionally burdened. Research requires systematicity and persistence in the search for a certain contingent.However, after collecting and processing the data, I have some concerns about the way the data is presented - the affected and unaffected limb of the patients are considered, but it should be taken into account that the affected limbs are not only left, or only right. Also, the authors mention that they measured porn response, but data from this are not presented. Despite my concerns after proper revision, the manuscript has potential.

6. PLOS authors have the option to publish the peer review history of their article (what does this mean? ). If published, this will include your full peer review and any attached files.

**Do you want your identity to be public for this peer review?** For information about this choice, including consent withdrawal, please see our Privacy Policy .

Reviewer #1: No

Reviewer #2: No

---

## [Author Response · Author response to Decision Letter 0]

14 Dec 2024

We have responded every comments of reviewers and editor in the attached file already.

---

## [Decision Letter · Decision Letter 1]

30 Dec 2024

PONE-D-24-38957R1Inter-lower limb and intra-lower limb muscle activity correlations during walking: A comparative study of stroke patients and healthy individualsPLOS ONE

Dear Dr. Chatpun,

Thank you for submitting your manuscript to PLOS ONE. After careful consideration, we feel that it has merit but does not fully meet PLOS ONE’s publication criteria as it currently stands. Therefore, we invite you to submit a revised version of the manuscript that addresses the points raised during the review process.

We look forward to receiving your revised manuscript.

Kind regards,

Tomoyoshi Komiyama, Ph.D

Academic Editor

PLOS ONE

Journal Requirements:

**Additional Editor Comments:**

Dear authors,

Thank you for your submitting your revised manuscript.

I think it is easier to understand than the previous revision.

However, two reviewers had additional questions.

Please answer these questions as listed below.

Tomoyoshi Komiyama

Reviewers' comments:

Reviewer's Responses to Questions

**Comments to the Author**

1. If the authors have adequately addressed your comments raised in a previous round of review and you feel that this manuscript is now acceptable for publication, you may indicate that here to bypass the “Comments to the Author” section, enter your conflict of interest statement in the “Confidential to Editor” section, and submit your "Accept" recommendation.

Reviewer #1: All comments have been addressed

Reviewer #2: (No Response)

2. Is the manuscript technically sound, and do the data support the conclusions?

Reviewer #1: Yes

Reviewer #2: Yes

3. Has the statistical analysis been performed appropriately and rigorously? 

Reviewer #1: Yes

Reviewer #2: Yes

4. Have the authors made all data underlying the findings in their manuscript fully available?

Reviewer #1: No

Reviewer #2: Yes

5. Is the manuscript presented in an intelligible fashion and written in standard English?

Reviewer #1: Yes

Reviewer #2: Yes

6. Review Comments to the Author

Reviewer #1: After reviewing the revised manuscript, I find that the authors have addressed most of my initial comments and the text has improved significantly. The manuscript has benefited from the revisions, particularly in the introduction and discussion sections. However, there are still some points that need further attention:

- Reference to SENIAM protocol:

The authors should include a proper reference to the SENIAM protocol utilized in the study. This is an important methodological detail.

- MATLAB mention:

The authors state that MATLAB2020 has been added to the text, but I could not find it in the revised manuscript. Please verify and ensure that it is included where appropriate.

- Future considerations using Vicon:

I believe it is essential to address the point I previously raised regarding the potential future use of Vicon systems to quantitatively validate the discussion points through kinematic data. This is crucial for strengthening the scientific rigor of the study.

- Figure captions (Figures 7 onward):

From Figure 7 onward, the captions should clarify the abbreviations used for each muscle. This will make the figures more accessible to readers.

- Figure quality:

The quality of the figures remains insufficient. When zooming in to examine details, the images become completely blurred. While the authors suggest that this issue stems from the PDF conversion tool, I recommend exploring alternative conversion programs to ensure clarity. Poor figure quality significantly hinders the understanding of the data.

- Table 2 and additional tables:

The presentation of Table 2 remains somewhat confusing, and now there are three separate tables for each affected side. I suggest unifying these tables if possible to improve readability. Additionally, for the healthy participants, the analysis does not consider dominance (right- or left-), which might be an important factor to address.

In conclusion, I believe the paper has become much stronger with the revisions made so far. It is well-supported and provides valuable insights. I recommend acceptance of the manuscript after the authors address the remaining points above.

Reviewer #2: A lot of work has been done on the article. Most of my comments have been taken into account. The authors could improve some details.

7. PLOS authors have the option to publish the peer review history of their article (what does this mean? ). If published, this will include your full peer review and any attached files.

**Do you want your identity to be public for this peer review?** For information about this choice, including consent withdrawal, please see our Privacy Policy .

Reviewer #1: No

Reviewer #2: No

---

## [Author Response · Author response to Decision Letter 1]

22 Jan 2025

Thank you for the comments. We have responded the comments and revised following the reviewers' suggestions. Please see our responses in the attached file.

---

## [Decision Letter · Decision Letter 2]

11 Feb 2025

Inter-lower limb and intra-lower limb muscle activity correlations during walking: A comparative study of stroke patients and healthy individuals

PONE-D-24-38957R2

Dear Dr. Chatpun,

We’re pleased to inform you that your manuscript has been judged scientifically suitable for publication and will be formally accepted for publication once it meets all outstanding technical requirements.

Kind regards,

Tomoyoshi Komiyama, Ph.D

Academic Editor

PLOS ONE

Additional Editor Comments (optional):

Dear authors,

Thank you for submitting your revised manuscript.

I think it was much easier to understand than the original manuscript.

I am satisfied with the responses and the edits, I am happy to accept this manuscript.

The authors have replied to my remaining comments satisfactorily from two reviewers.

Therefore, I have no further comments to make, all of my previous concerns were adequately addressed. This manuscript will be satiating the reader's interest.

Tomoyoshi Komiyama

Reviewers' comments:

Reviewer's Responses to Questions

**Comments to the Author**

1. If the authors have adequately addressed your comments raised in a previous round of review and you feel that this manuscript is now acceptable for publication, you may indicate that here to bypass the “Comments to the Author” section, enter your conflict of interest statement in the “Confidential to Editor” section, and submit your "Accept" recommendation.

Reviewer #1: All comments have been addressed

Reviewer #2: All comments have been addressed

2. Is the manuscript technically sound, and do the data support the conclusions?

Reviewer #1: Yes

Reviewer #2: Yes

3. Has the statistical analysis been performed appropriately and rigorously? 

Reviewer #1: Yes

Reviewer #2: Yes

4. Have the authors made all data underlying the findings in their manuscript fully available?

Reviewer #1: No

Reviewer #2: Yes

5. Is the manuscript presented in an intelligible fashion and written in standard English?

Reviewer #1: Yes

Reviewer #2: Yes

6. Review Comments to the Author

Reviewer #1: (No Response)

Reviewer #2: After the authors have significantly improved their manuscript and accommodated most of the reviewers' recommendations, I recommend this article for publication.

7. PLOS authors have the option to publish the peer review history of their article (what does this mean? ). If published, this will include your full peer review and any attached files.

**Do you want your identity to be public for this peer review?** For information about this choice, including consent withdrawal, please see our Privacy Policy .

Reviewer #1: No

Reviewer #2: No

---

## [Editor Report · Acceptance letter]

PONE-D-24-38957R2

PLOS ONE

Dear Dr. Chatpun,

I'm pleased to inform you that your manuscript has been deemed suitable for publication in PLOS ONE. Congratulations! Your manuscript is now being handed over to our production team.

Kind regards,

on behalf of

Dr. Tomoyoshi Komiyama

Academic Editor

PLOS ONE